

# CNN-based noise reduction for multi-channel speech enhancement system with discrete wavelet transform (DWT) preprocessing

Pavani Cherukuru[1,2] and Mumtaz Begum Mustafa[1]

[1] Department of Software Engineering, Faculty of Computer Science and Information Technology, Universiti Malaya, Kuala Lumpur, Malaysia
[2] Department of Information Science, Dayananda Sagar Academy of Technology and Management, Bangalore, Karnataka, India

## ABSTRACT

Speech enhancement algorithms are applied in multiple levels of enhancement to improve the quality of speech signals under noisy environments known as multi-channel speech enhancement (MCSE) systems. Numerous existing algorithms are used to filter noise in speech enhancement systems, which are typically employed as a pre-processor to reduce noise and improve speech quality. They may, however, be limited in performing well under low signal-to-noise ratio (SNR) situations. The speech devices are exposed to all kinds of environmental noises which may go up to a high-level frequency of noises. The objective of this research is to conduct a noise reduction experiment for a multi-channel speech enhancement (MCSE) system in stationary and non-stationary environmental noisy situations with varying speech signal SNR levels. The experiments examined the performance of the existing and the proposed MCSE systems for environmental noises in filtering low to high SNRs environmental noises (−10 dB to 20 dB). The experiments were conducted using the AURORA and LibriSpeech datasets, which consist of different types of environmental noises. The existing MCSE (BAV-MCSE) makes use of beamforming, adaptive noise reduction and voice activity detection algorithms (BAV) to filter the noises from speech signals. The proposed MCSE (DWT-CNN-MCSE) system was developed based on discrete wavelet transform (DWT) preprocessing and convolution neural network (CNN) for denoising the input noisy speech signals to improve the performance accuracy. The performance of the existing BAV-MCSE and the proposed DWT-CNN-MCSE were measured using spectrogram analysis and word recognition rate (WRR). It was identified that the existing BAV-MCSE reported the highest WRR at 93.77% for a high SNR (at 20 dB) and 5.64% on average for a low SNR (at −10 dB) for different noises. The proposed DWT-CNN-MCSE system has proven to perform well at a low SNR with WRR of 70.55% and the highest improvement (64.91% WRR) at −10 dB SNR.

Corresponding author
Mumtaz Begum Mustafa,
mumtaz@um.edu.my

# INTRODUCTION

The speech enhancement system reduces background disturbances/noises while protecting against any changes to speech features to deal with noisy speech signals. Speech enhancement distinguishes between the intended speech and background noise interference (*Wang & Chen, 2018*). It aims to enhance speech quality to optimize associated signal processing systems, such as wearables (*Takada, Seki & Toda, 2018*), automatic speech recognition (*Donahue, Li & Prabhavalkar, 2018*), mobile telephony (*Hasannezhad et al., 2021*), and hearing prostheses (*Syed, Trinh & Mandel, 2018*). Many algorithms have been proposed, and the noise issue has been studied extensively for a very long period (*Das et al., 2020*). Spectral subtraction algorithms (*Balaji et al., 2020*), Wiener filtering (*Yang & Bao, 2018*), and nonnegative matrix factorization (*Xu et al., 2021*) are examples of traditional speech enhancement algorithms. However, only some of these algorithms pay attention to speech enhancement at low signal-to-noise ratio (SNR) conditions, which is more important and challenging than high SNR conditions. Generally, −10 dB to 0 dB SNR levels refer to low SNR's, while 5 dB to 15 dB are high-level SNR's (*Wang & Chen, 2018*). There are many communication scenarios at low SNR conditions. For instance, walkie-talkies used by employees in metal-cutting factories, wireless headsets used by mechanics when testing a helicopter, and so on. The current focus of research is on improving the performance of communication devices such as microphones, automatic speech recognition (ASR), Voice over Internet Protocol (VoIP), teleconferencing *etc.*

While speech enhancement at high SNR makes the speech more comfortable for listeners, speech enhancement at low SNR affects the clarity of the speech. The improvement of speech at low SNR is not therefore more significant than at high SNR (*Xu et al., 2004*). However, dealing with a high noise level in a noisy environment and providing noise-free communication is a trending research topic in this field. Several algorithms, such as spectral subtraction, beamforming, adaptive noise reduction, spectral statistical filter, among others, have been proposed to improve speech quality.

Multi-channel speech enhancement (MCSE) refers to systems that make use of multiple signal inputs, use noise references in adaptive noise cancellation, phase adjustment to cancel unwanted noise components, and combine step-by-step schemes (*Kokkinakis & Loizou, 2010*). The existing MCSE provides speech recognition at a 71% Word Recognition Rate (WRR) at 10 dB SNR compared to a single microphone (*Xu et al., 2004*; *Stupakov et al., 2012*). These multi-channel algorithms (beamforming, adaptive noise reduction and voice activity detection algorithms) suffer from the low performance of recognition rate when SNR is low (−15 dB, −10 dB, −5 dB, 0 dB) (*Pauline, Samiappan & Kumar, 2021*; *Kim, 2020*). The existing algorithms developed for MCSE systems were only tested for white Gaussian stationary noise at 0 to 60 dB SNRs and were never tested for non-stationary environmental noises.

The deep learning algorithm is one of the state-of-the-art algorithms in the speech enhancement domain (*Rownicka, Bell & Renals, 2020*; *Ochiai, Delcroix & Nakatani, 2020*), which has been proven to have acceptable performance in handling different levels of noise in speech enhancement based on the computing platform. Among the deep learning

algorithms, the very deep convolution neural network (VDCNN-conv) reported the highest WRR at 90.45% and an average WRR of 87.45% for environmental noises (*Cherukuru, Mumtaz Begum & Hema, 2021*). However, MCSE systems have never been experimented with deep learning algorithms. As such, the aim of this research is to propose a MCSE system using deep learning and preprocessing algorithms and examine the performance of the proposed system against the existing MCSE system in filtering environmental noises at low to high SNR conditions.

The rest of the article is structured as follows. The 'Research background' provides an overview of the single-channel and multi-channel speech enhancements and their limitations which include the existing MCSE and deep learning algorithms. The next section describes the proposed deep learning based MCSE system. The 'Methodology' section describes the approach, experimental design, setup, and evaluation methods used on both the benchmark MCSE speech enhancement and the proposed approach of MCSE. The 'Results' section presents the findings of this research, while discussions are presented in the next section. Finally, the last section concludes the proposed work.

## RESEARCH BACKGROUND

Many existing algorithms were used to filter noise in MCSE systems and are often used as a pre-processor to improve speech quality. They have proven to be effective in reducing interference signals and improving voice quality. There are two categories of speech enhancement systems, which are single-channel and multi-channel speech enhancement.

### Single channel speech enhancement system

The approaches for enhancing speech with only one acquisition channel are known as "single channel" algorithms. A single channel is typically not available in most real-time applications such as speaker recognition, voice recognition, mobile communications, and hearing aids, though they are relatively cheaper than multi- channel systems. This is one of the most difficult situations in speech enhancement domain as there is no reference signal available for noise, and clean speech/audio signal cannot be preprocessed before it is affected by the noise (*Yadava & Jayanna, 2019*; *Hossain et al., 2023*; *Xu, Tu & Yang, 2023*). Despite the challenges, there are several algorithms developed and experimented such as subtraction algorithms, over subtraction algorithms, non-linear spectral subtraction, non-linear weighted subtraction, *etc*. These algorithms improved the performance of speech quality in noisy environments; however, they're computationally intensive and not effective at suppressing noisy audio signals, especially when the SNR is low *i.e.,* $-10$ dB to 10 dB (*Shanmugapriya & Chandra, 2014*; *Upadhyay & Karmakar, 2015*; *Saleem et al., 2022*). This environmental noise is difficult to filter because it has different characteristics in terms of noisy levels in decibels, frequencies *etc*. depending on the type of environment. Therefore, MCSE is very much required (*Akhaee, Ameri & Marvasti, 2005*).

Recently, researchers have given attention to the convolutional neural network (CNN) algorithm for single-channel speech enhancement system. The performance of CNN was measured using various measurement metrics such as mean opinion score (MOS), signal

distortion (SIG), and intrusiveness of background noise (BAK). A scale from 1 to 5 is used for SIG, BAK, and MOS, with a higher number being preferred.

While word error rate (WER) or word recognition rate (WRR) is a common metric used to specifically assess the performance of ASR systems, other common objective measures include segmental signal-to-noise ratio (segSNR), distance measures, source-to-distortion ratio (SDR), perceptual evaluation of speech quality (PESQ), and short-time objective intelligibility (STOI).

In *Soleymanpour et al. (2023)*, speech enhancement in a single channel was implemented using CNN algorithms for complex noisy speeches to improve the speech quality (*Passricha & Aggarwal, 2019*) which produces the following result; PESQ = 3.24 (*Wang & Wang, 2019*; *Park & Lee, 2017*), CSIG (signal distortion) = 4.34 (*Pandey & Wang, 2019*; *Germain, Chen & Koltun, 2019*), CBAK (background noise interference) = 4.10 (*Fu et al., 2018*; *Rownicka, Bell & Renals, 2020*), COVL (overall quality of speech) = 3.81 (*Rethage, Pons & Serra, 2018*), and SSNR (Segmented Signal to Noise Ratio) = 16.85 (*Choi et al., 2019*). Additionally, CNN was said to be more effective than recursive neural networks (RNNs) (*Park & Lee, 2017*) and traditional feedforward neural networks (*Oord et al., 2016*). According to *Park & Lee (2017)*, CNN can perform better with a network that is 12 times smaller than RNN (*Park & Lee, 2017*). CNN is effective in distinguishing the speech and noise components of noisy signals because it can handle the local temporal spectral features of speech. Both in the spectrum and waveform domains, CNN has demonstrated its efficacy in improving speech.

## Multi-channel speech enhancement system (MCSE)

Microphone arrays and speech enhancement components are built into MCSE that processes multiple channels of audio signals in noisy environments such as outdoor environments (*Palla et al., 2017*; *Pauline, Samiappan & Kumar, 2021*). For example, a spectral statistics filter is applied to hearing aids to handle stationary noise environments (Gaussian noise) and unsteady noise environments (factories, babble, and car noises) from −5 dB to 20 dB (*Kim, 2020*). The current performance rate at low SNR are 2.16 PESQ score with babble noise, 2.20 considered as low quality of signal with Gaussian noise, 2.13 considered as low quality of signal with factory noise and 3.67 PESQ score considered as a medium quality of signal with car noise on an average of −5 dB to 10 db SNR levels (*Kim, 2020*).

Figure 1 shows the architecture of the existing MCSE system based on beamforming, adaptive noise reduction, and voice activity detection (BAV-MCSE). The architecture consists of a microphone array, beamforming, adaptive noise reduction, and voice activity detection.

- Beamformer

A microphone array can be combined with a spatial filtering signal processor called Beamformer. Beamforming is achieved with filtering the microphone signal, merging the outputs to obtain the desired signal, and filtering away interference noise (*Van Veen & Buckley, 1988*). Fixed beamforming and adaptive beamforming are the two types of beamforming. The direction of the input signal is fixed in fixed beamforming, and the

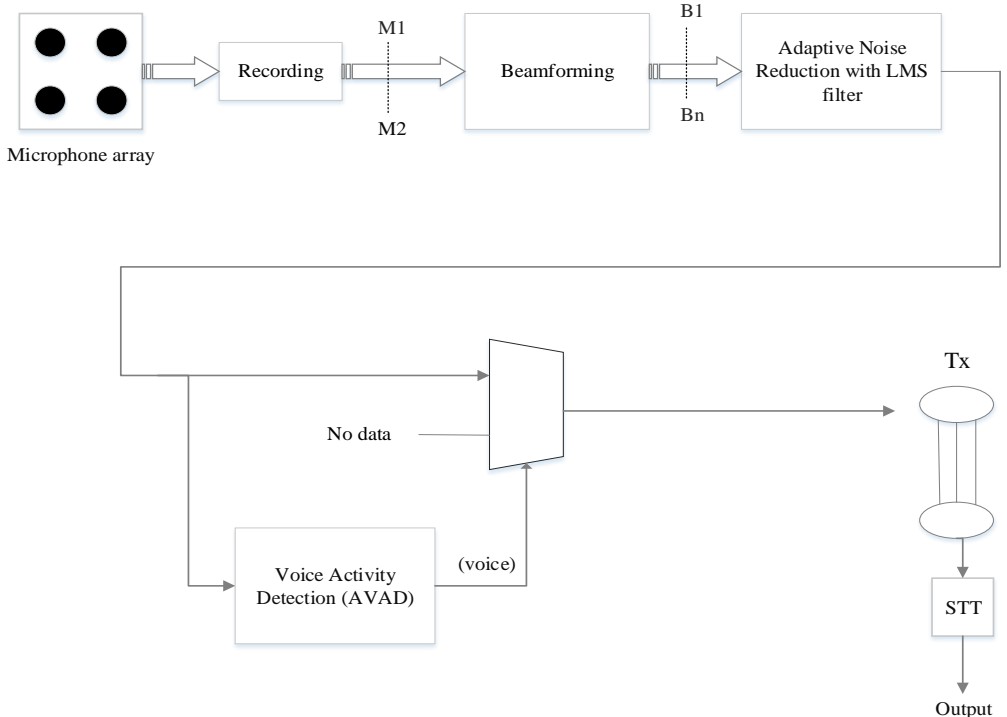

**Figure 1** **Architecture of the existing multi-channel speech enhancement (BAV-MCSE).**

distance between the microphones is constant. Fixed beamforming is achievable with delay and sum beamformers. In adaptive beamforming, the directivity of the input noisy speech signal varies as the acoustic environment changes (*Ramesh Babu & Sridhar, 2021*).

• Adaptive Noise Reduction (ANR)

The Least mean square (LMS) filter is used to filter environmental noise using Adaptive Noise Reduction (ANR) (*Soo & Pang, 1990*; *Valin, 2007*). The ANR is fed by the user beam and reference noise (b1...bn). The ANR component filters the noise from the user beam, which is connected to the reference noise. However, the audio signal is only present in the user beam after it has been processed using beamforming but is not attenuated (*Widrow et al., 1975*).

• Voice Activity Detection algorithm (VAD)

VAD distinguishes the user's voice in the user stream (*Karita et al., 2019*), which is important for two reasons. The first one is for segmentation, where the system identifies the precise boundaries of each word in spoken utterances. The second one is for data reduction, where the system only sends data when it is needed, rather than continuously transmitting data through the transmission channel.

According to *Cherukuru, Mumtaz Begum & Hema (2021)* only white Gaussian stationary noise was tested with beamforming, ANR and VAD algorithms in MCSE between 0 dB to 60 dB SNRs, and proving that MCSE is particularly effective at 20 dB SNR (*Palla et al., 2017*). For the stationary white Gaussian noise, the word recognition rate decreased for

low noises at 15 dB to −10 dB, respectively (_Cherukuru, Mumtaz Begum & Hema, 2021_). The negative dB had a lower WRR than the positive dB but low for 15 dB to −10 dB non-stationary noise. At low SNR conditions, the existing MCSE's WRR was poor. It did, however, perform better in a noisy stationary environment than in a noisy non-stationary environment.

## Deep learning-based multi-channel speech enhancement system

Deep neural networks (DNNs) is the first introduced technology for guided speech improvement, have grown in prominence in recent years (_Venkatesha Prasad et al., 2002_). DNN, also known as the feed-forward fully connected layer or multilayer perception (MLP) with numerous hidden layers, is one of the most used designs for speech enhancement (_Zhao et al., 2018_). The network is characterised as a completely connected network because every node in the layer has a link with every node in the layer preceding it. Resultantly, DNN contains many parameters.

_Karjol, Kumar & Ghosh (2018)_ introduced an enhancement strategy based on numerous DNN-based systems with n number of DNN, each of which contributes to the final enhanced speech, and a gating network that provides weights to combine the DNN outputs. The standard metrics can be used to compare the performance of speech enhancement systems using subjective and objective measures. The model employs $n = 4$, with each layer consisting of three layers. An average SNR of −5 dB to 10 dB on the TIMIT corpus, yields a seen noise PESQ of 2.65 and an unseen noise PESQ of 2.19.

Although DNN has been used successfully as a regression model for speech enhancement, its improved speech frequently degrades in low SNR conditions (_Gao et al., 2016_). To enhance the effectiveness of DNN-based speech in low SNR environments, some scholars presented a progressive learning architecture with long short-term memory (LSTM) network (_Gao et al., 2018_; _Santhanavijayan, Kumar & Deepak, 2021_). Each target layer is built so that the transition speech with a higher SNR is learned at the final layer, followed by clean speech. Additionally, LSTM-RNN has been used to solve the issues with reverberation (_Weninger et al., 2013_), loud multichannel speech and extremely non-stationary additive noise (_Wollmer et al., 2013_). In _Wollmer et al. (2013)_, bottleneck features produced by the bi-directional LSTM network (BiLSTM) outperformed manually created features like MFCC. When employing MFCC, the average word accuracy (WA) is 38.13%, whereas when using batch-normalized long short-term memory (BN-BLSTM), it is 43.55%. The LSTM-RNN has significantly enhanced speech processing systems. However, it is well known that learning the RNN parameters is challenging and time-consuming (_Weninger et al., 2013_; _Wollmer et al., 2013_).

This research proposed a noise-reduction framework using pre-processing and deep learning algorithms to overcome the noise issue in MCSE system. Based on _Katti & Anusuya (2011)_, _Labied & Belangour (2021)_, _Ping, Li-Zhen & Dong-Feng (2009)_, discrete wavelet transform (DWT) preprocessing algorithm and CNN algorithm are suitable for filtering noisy environments and improving the quality of speech. According to _Labied & Belangour (2021)_, DWT is effective in denoising speech signal, and can compress the speech signal without degrading the speech quality. In _Katti & Anusuya (2011)_, _Ping,_

*Li-Zhen & Dong-Feng (2009)*, it was stated that CNN has the capacity to detect patterns in neighboring speech structures, and compared to RNN and standard DNN, CNN is more effective in terms of filtering the high level of noise in speech signals. However, it is unable to maintain invariance when the input data changes. Among all the deep learning algorithms, CNN reported the highest WRR at 90.45% and the lowest WRR at 87.45% on average for environmental noises.

Table 1 summarizes the deep learning-based algorithms for single and multi-channel speech enhancement. CNN with preprocessing algorithms is yet to be experimented with the MCSE system to improve speech quality mainly under low SNR conditions.

# THE PROPOSED CNN-BASED NOISE REDUCTION FOR MULTI-CHANNEL SPEECH ENHANCEMENT SYSTEM WITH DWT PRE-PROCESSING

This section presents the proposed architecture for an MCSE system by proposing the DWT preprocessing and a CNN-based deep learning algorithm (DWT-CNN-MCSE). DWT algorithm is used as a pre-processing technique to remove selected noise by decomposing the noisy speech signals. CNN algorithm is used to handle feature extraction and classification. The proposed architecture is depicted in Fig. 2.

### Discrete wavelet transform (DWT) preprocessing

This research adopted DWT among other algorithms as its performance is very effective in terms of denoising the speech signal and compressing the speech signal without any significant loss in speech quality (*Katti & Anusuya, 2011*; *Labied & Belangour, 2021*; *Ping, Li-Zhen & Dong-Feng, 2009*).

This algorithm aims to create by rescaling and iterating through a series of filters. Up-sampling and down-sampling (subsampling) processes determine the signal's resolution (detail information), whereas filtering operations determine its scale (resolution). As there is a lack of preprocessing algorithms implemented on MCSE systems, this article used the existing DWT preprocessing algorithm to remove the redundant data from noisy speech signals.

This research implements a discrete wavelet-based algorithm for the signals obtained through MEMS microphones. Algorithm 1 explains the step-by-step procedure of DWT applied in this research. To ensure the wavelet series is properly computed, which is a sampled form of continuous wavelet transform (CWT), it may take a significant amount of time and resources. There is evidence that the sub-band coding-based DWT is more efficient in computing wavelet transforms. It is simple to implement, and it decreases the time and resources needed for computation. Digital filtering algorithms are used to obtain a time-scale depiction of the digital signal in DWT. Filters with various cutoff frequencies and scales are used to evaluate the input signal.

### Convolutional Neural Network (CNN)

This research adopted CNN (*Katti & Anusuya, 2011*; *Ping, Li-Zhen & Dong-Feng, 2009*) as it is effective in distinguishing between the speech and noise components of noisy

**Table 1** Summary of deep learning-based algorithms used for single-channel and multi-channel speech enhancement systems in filtering different types of noises.

| Deep learning method | References | Dataset | Evaluation metrics | Results | Advantages/Disadvantages |
|---|---|---|---|---|---|
| DNN (Deep Neural Network) | *Zhao et al. (2018)* | NOISEX and IEEE corpus | SDR, PESQ, and STOI | Averaged results with mismatched SNR (−3 to 3 dB) PESQ is 1.99, SDR is 11.35, and STOI is 90.61%. | **Advantages** Being familiar with the model's architecture since Networks are typically simple. **Disadvantages** DNN has relatively big parameters since every node in each layer is connected to every node in the layer before it. |
| | *Karjol, Kumar & Ghosh (2018)* | TIMIT + noises from AURORA dataset | STOI, SegSNR,and PESQ | For seen noise, the average best PESQ is 2.65, whereas for unseen noise, it is 2.19. | |
| | *Saleem & Khattak (2020)* | Environmental noises | SegSNR, PESQ, LLR and STOI | PESQ is 2.27, SNRseg is 4.24 , LLR is 0.53 and STOI is 84% | |
| Deep autoencoder based on MFCC (DAE-MFCC) | *Feng, Zhang & Glass (2014)* | CHiME-2 | WER | Error rate of 34%. | **Advantages** Dimensional reduction is done using DAE, and the bottleneck layer's features might be helpful. **Disadvantages** Learning temporal information is a drawback of DNN-based DAE information. |
| | *Lu et al., (2013)* | Japanese corpus + environmental noises | PESQ | Average PESQ for factory noise is 3.13, whereas it is 4.08 for car noise. | |
| Recurrent neural network-Long short-term memory (RNN-LSTM) | *Gao et al. (2018)* | In factories, the average PESQ is 3.13, and in cars, it is 4.08. | SDR, STOI | STOI: 0.86 and SDR: 9.46 on average. | **Advantages** -Best for handling data that is sequence-based, like speech signals. -Contextual data can be handled by RNN-LSTM. **Disadvantages** It is well known that learning the RNN parameters is challenging and time-consuming. |
| | *Weninger et al. (2013)* | CHiME-2 | WA, WER | Average accuracy is 85%. | |
| | *Wollmer et al. (2013)* | Buckeye (spontaneous speech) + CHiME noises | WA | Average WA using BN-BLSTM: 43.55%. | |
| | *Maas et al. (2012)* | AURORA-2 | MSE and WER | The average error rate (SNR 0-20 dB) is 10.28% for seen noise and 12.90% for unseen noise. | |

| Deep learning method | References | Dataset | Evaluation metrics | Results | Advantages/Disadvantages |
|---|---|---|---|---|---|
| | *Wang & Wang (2019)* | CHiME-2 + environmental Noises | WER | Magnitude features provide the best average error rate of 7.8% (accuracy of 92.2%). | |
| | *Park & Lee (2017)* | TIMIT + environmental noises | PESQ, STOI, SDR | CNN outperformed DNN and RNN in terms of accuracy, with PESQ 2.34, STOI 0.83, and SDR 8.62. | |
| | *Plantinga, Bagchi & Fosler-Lussier (2019)* | CHiME-2 | Word Error Rate (WER) | Using ResNet and mimic loss, a word error rate of 9.3% is achieved. | |
| | *Rownicka, Bell & Renals (2020)* | AMI and Aurora-4 | Word Error Rate (WER) | 8.31% WER on Aurora-4 | |
| | *Pandey & Wang (2019)* | NOISEX + TIMIT + SSN | STOI, PESQ, and SI-SDR | Results indicate that Autoencoder CNN performed better than SEGAN. | |
| | *Germain, Chen & Koltun (2019)* | Voice Bank + DEMAND | SNR, SIG, BAK, OVL | SNR:19.00, SIG: 3.86, BAK: 3.33, OVL: 3.22. | |
| | *Fu et al. (2018)* | TIMIT + environmental noises | PESQ, STOI | Fully utilising ConvNet yields the best STOI, while DNN achieves the best PESQ. | |
| | *Donahue, Li & Prabhavalkar (2018)* | WSJ + environmental and Music noise | Word Error Rate (WER) | 17.6% word error rate. | |
| | *Baby & Verhulst (2019)* | Voice Bank + DEMAND | STOI, PESQ, SegSNR | PESQ: 2.62, SegSNR: 17.68, STOI: 0.942 | |
| | *Ochiai, Delcroix & Nakatani (2020)* | CHiME-4, Aurora-4 | WER, SDR | Chime-4: SDR: 14.24, Aurora-4: 6.3%, WER: 8.3% (real data), 10.8% (simulated). | **Advantages** -CNN has the capacity to detect patterns in neighbouring speech structures. -Compared to RNN and standard DNN, CNN is more effective. **Disadvantages** Inability to maintain invariance when the input data changes |
| | *Xu, Elshamy & Fingscheidt (2020)* | Grid corpus + CHiME-3 noises | PESQ, STOI | For seen noises, PESQ is 2.60 and STOI is 0.70, while for unseen noises only, 2.63 and 0.74. | |

**Table 1** (*continued*)

| Deep learning method | References | Dataset | Evaluation metrics | Results | Advantages/Disadvantages |
|---|---|---|---|---|---|
| | *Choi et al. (2019)* | Voice Bank + DEMAND | PESQ, CSIG, CBAK, COVL, SSNR | PESQ 3.24, CSIG 4.34, CBAK 4.10, COVL 3.81, and SSNR 16.85 are the values. | |
| CNN (Convolution neural network) | *Soleymanpour et al. (2023)* | Babble Noise | PESQ, STOI | PESQ is 1.35 to 1.78 at -8db to 0db and STOI is 0.56 | |
| | *Saleem et al. (2023)* | VoiceBank-DEMAND Corpus + Librispeech | PESQ, STOI. | PESQ is 2.28, STOI is 84.5%. | |
| GAN (generative adversarial network) | *Soni, Shah & Patil (2018)* | Voice Bank + DEMAND | PESQ, CSIG, CBAK, MOS, STOI | PESQ 2.53, SIG 3.80, BAK 3.12, MOS 3.14, and STOI 0.93T are the values. | **Advantages:** If GAN is correctly trained, its combined networks can be very strong. **Disadvantages:** The adversarial training is typically challenging and unstable. |

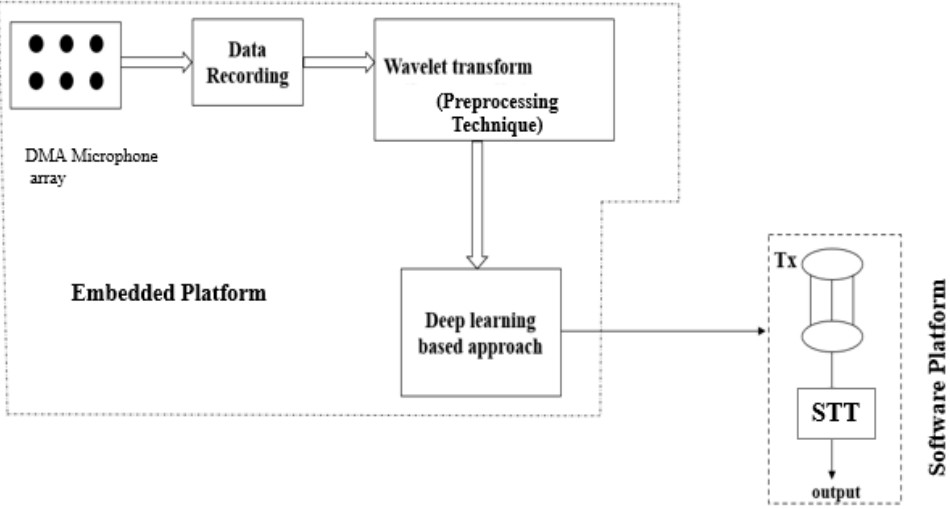

**Figure 2** **The proposed deep learning-based multi-channel speech enhancement (DWT-CNN MCSE) architecture using wavelet transform preprocessing technique.**

signals due to its ability to handle the local temporal spectral features of speech. Both in the spectrum and waveform domains, CNN has demonstrated its efficacy in improving speech recognition rate. Additionally in this research, the extended CNN with BiLSTM layer before it applies to fully connected layer is used.

Even though CNNs effectively simulate the structural locality from the feature space, as CNN adopts the pooling at a limited frequency domain, it reduces the linear variance and handles disturbances and minor shifts in the feature space. By making use of prior

**Algorithm 1**: Preprocessing the input signal using Discrete Wavelet Transform algorithm

Input: original noisy speech signal, wavelet decomposition bands Output: decomposed signals and corresponding coefficients

Xdata[] stores the input data vector, and Ydata[ ] is the output data vector that is returned. N is the length of both data vectors. Before applying this approach, it is presumable that the wavelet filter parameters G[k] and the scale filter parameters H[k] have been provided. L is the total number of parameters. N must be an even number to work with this algorithm.

**Step 1:** Set $s = \frac{N}{2}$ // Start index of the input array's gamma coefficients
**Step 2:** Allocate $ydata[N]$; // Provide a memory space for the output data vector

**Step 3:** for $(i = 0 \ while \ i < N \ increment \ i = i+1)$ do // loop over input data
**Step 4:** $ydata[i] = 0$; // Reset summation accumulators.
**Step 5:** endfor;
**Step 6:** $j = 0$; // access/index to the output data array
**Step 7:** for $(i = 0 \ while \ i < N \ increment \ i = i+1)$ do // loop over input data.
**Step 8:** for $k = 0 \ while \ k < L \ increment \ k = k+1)$ do // convolution loop.
**Step 9:** $didx = (i+k) \, mod \, N$; // access/index into input data with wraparound.
**Step 10:** $ydata[j] = ydata[i] + G[k] * xddata[didx]$; // Scaling filter contribution

**Step 11:** $ydata[j+s] = ydata[i+s] + H[k] * xddata[didx]$; // Wavelet filter contribution

**Step 12:** endfor;
**Step 13:** $j = j+1$; // Update position in output array
**Step 14:** endfor;

knowledge of the speech signal, they can take advantage of the long-term dependencies between the speech frames. However, CNNs in speech communication systems cannot handle many semi-clean data, resulting in reduced performance. To overcome these issues bidirectional long short-term memory (BiLSTM), which regulates the flow of information by an individual component called a memory block, was developed by *Weninger et al. (2013)*.

The fundamental purpose of CNN is to detect local structure in input data. The spectrum correlations in acoustic features are well-modeled by CNN, which successfully decreases the spectral fluctuations. Three distinct models such as CNN, BiLSTM, and fully connected layers are included in the suggested architecture as illustrated in Fig. 3.

Convolutional layers are used to reduce the frequency variance in the input signal at first. Two CNN layers with 256 feature mappings in each convolutional layer were chosen at first. This is because speech has a tiny feature dimension (*i.e.,* 40). The behavior of the high- and low-frequency zones is vastly different. Nearly 16% of the feature map's

**Figure 3** CNN-BLSTM architecture applied for multi-channel speech enhancement system.

original size has been decreased using two convolutional layers. As a result, modeling locality and eliminating invariance is no longer necessary. *Sainath et al. (2013)* states that the first convolutional layer has a 9 by 9 frequency-time filter, while the second layer has a 4 by 3 frequency-time filter. A 9 by 9 frequency-time filter is used in the first convolution layer, and a 4 by 3 frequency-time filter is used in the second. In the beginning, our model employs solely the frequency-domain pooling using max pooling. Similarly, the pooling size is 2 for both layers, and the stride value is 2. The next layer in CNN has a greater dimension since the set of feature maps, time, and frequency is proportional to the layer's size. Therefore, the feature dimensions must be reduced. After CNN layers, a linear layer is applied to reduce the layer's size without sacrificing accuracy.

Algorithm 2 shows the processes involved in CNN through different layers. Frequency modeling is an algorithm for reducing the data dimensionality of 236 suitable outputs by using linear layers. To simulate the signal in time, the output of the CNN layer is passed to the BiLSTM layer. In this case, two BiLSTM and three FC layers would be ideal. However, the number of layers can vary depending on the experiment. Each BiLSTM layer has 832 cells and 512 units (256 LSTM units per direction) of the projection layer for feature extraction (256 LSTM units). Twenty-times steps are pre-trained into the BiLSTM, and backpropagation is truncated. The output of BiLSTM layers is sent to FC layers after frequency and timing modeling. Higher-order feature representations that are easily distinguishable between classes can be generated by using these layers. 1024 hidden units can be found in all fully connected layers.

Speech variation results from the accent, volume, and other characteristics can distinguish distinct speakers. The proposed approach uses shared weights obtained by applying several convolution operations. These convolutions generate features and are supplied to the Max pooling layer. The shared weights mechanism helps retain the top-level and low-level attributes as well as improve the accuracy in terms of WRR. Further, these attributes are processed through the Linear Layer, which supplies these features to CNN-BiLSTM layer (*Sermanet, Chintala & LeCun, 2012*; *Passricha & Aggarwal, 2019*). In most CNN work, FC layers discriminate between classes based on local knowledge. CNN-BiLSTM module is used for energy and timing modeling, and the softmax layer is

utilized to distinguish between different classes. The entire model is trained at the same time.

---

**Algorithm 2:** Processing DWT output signals through CNN-BLSTM Algorithm
Input: speech signals, Deep learning parameter (batch size, feature dimension, classes, train test ratio).

Output: enhanced speech signal with recognition rate performance.

**Step 1:** capture speech signals by using DMA microphone array

**Step 2:** Apply an analogue to digital converter to convert an analogue signal into a digital signal.

**Step 3:** apply wavelet transform by applying $X(a, b) = \frac{1}{\sqrt{a}} \int_{-\infty}^{\infty} \psi\left(\frac{t-b}{b}\right) x(t) \, dt$
- Decompose signal into LL, HL, LH, and HH bands by computing the wavelet coefficients as $c_{jk} = \left[W_\psi f\right]\left(2^{-j}, k2^{-j}\right)$

**Step 4:** Input these coefficients to deep learning
- Process through convolutional layers $n_{out} = \left[\frac{n_{in} + 2p - k}{s}\right] + 1$, $n_{in}$ denotes the input attributes, $n_{out}$ denotes the output features, $k$ convolution kernel size, $p$ padding size, $s$ is the stride

- Process the convolved data through pooling layer $h_{xy}^l = \max_{i=0,\ldots,s, j=0,\ldots,s} h_{(x+1)(y+j)}^{l-1}$

- Perform linearization by applying linear layer
- Apply BiLSTM layer
- Process the memory unit data through fully connected layer $z^l = W^l h^{l-1}$
- Soft max layer $softmax(z_i) = \frac{e^{z_i}}{\sum_j e^{z_j}}$

**Step 5:** obtain the final output speech data and measure the performance

---

## RESEARCH METHOD

This research aims to examine the performance of the proposed MCSE systems in filtering stationary and non-stationary environmental noises with low to high SNRs environmental noises ($-10$ dB to 20 dB). The experiments compare the ability of the proposed DWT-CNN-MCSE system against the existing BAV-MCSE in filtering the noise environment at low SNR conditions. The existing and the proposed MCSE systems were evaluated in terms of spectrogram analysis and WRR.

### Experimental design

The experimental design of this research was based on the researchers' previous work *Cherukuru, Mumtaz Begum & Hema (2021)*, which was for environmental noises at different levels of SNRs to determine the limitations of the existing algorithms in handling environmental noises. From our previous work (*Cherukuru, Mumtaz Begum & Hema, 2021*), we found that the existing MCSE shows an acceptable recognition rate at high SNR levels but not for low SNR levels. To overcome the problem of low recognition rate

**Table 2  Experimental design.**

| Technique | Techniques used in the multi-channel speech enhancement system | Speech database | SNR/dB | Noises | |
|---|---|---|---|---|---|
| The existing BAV-MCSE | Beamforming, ANR and VAD | AURORA | −10 dB, −5 dB, 0 dB, 5 dB, 10 dB, 15 dB | **Stationary Noise:** White Gaussian noise | **Non-stationary Noises:** Airport, Babble, Car, Exhibition, Restaurant |
| Proposed DWT-CNN-MCSE | Convolution neural network (CNN) and discrete wavelet feature extraction (DWT) technique | AURORA | −10 dB, −5 dB, 0 dB, 5 dB, 10 dB, 15 dB | **Stationary Noise:** White Gaussian noise | **Non-stationary Noises:** Airport, Babble, Car, Exhibition, Restaurant |
| | | LibriSpeech | −10 dB, −5 dB, 0 dB, 5 dB, 10 dB, 15 dB | N/A | Dog bark noise, Fan noise, Ambulance noise |

for low SNR levels, this research proposed a deep learning-based algorithms to improve the recognition accuracy of the MCSE system. The experimental design of the proposed DWT-CNN-MCSE system is shown in Table 2. The experiment adds noise to the original signals at levels of −10 dB, −5 dB, 0 dB, 5 dB, 10 dB, and 15 dB before processing them through the considered MCSE system.

## Speech dataset

The AURORA and the LibriSpeech datasets were used to train the deep learning models to test the MCSE systems (benchmark and the proposed systems) in noisy environments at various SNR levels of voice signals.

- AURORA

There were 13 distinct male voices and 16 distinct female voices among the 25 utterances taken from the AURORA noisy sample (*Karjol, Kumar & Ghosh, 2018*). Even though the number of trials changes based on the noise level, at least 25 samples were taken for each dB level.

Five types of non-stationary environmental noise are represented in the noisy speech utterances, which are airport, babble, exhibition, car, and restaurant noises. One stationary noise type, white Gaussian noise, was examined at seven different SNRs: −10 dB, −5 dB, 0 dB, 5 dB, 10 dB, 15 dB, and 20 dB. Other noise types that were investigated included babble, car, exhibition, and restaurant noises, while −10 dB was for loud speech signals. A total of 25 utterances from the AURORA clean training dataset were selected, and −10 dB noise signals were purposefully mixed with it. For every 25 utterances, 42 different conditions were prepared.

In the proposed experiment, the AURORA database, which is taken from the internationally recognized NOIZEUS database for the evaluation of speech enhancement algorithms was used. This database includes the speech recordings of speakers, three men,

and three women, reciting 30 sentences from the IEEE sentence database. The University of Texas at Dallas' Speech Processing Lab used Tucker Davis Technology (TDT) to capture each speaker's five words at a sampling frequency of 25 kHz, which was later down sampled to 8 kHz. Every sentence is accompanied by a variety of background noises, including those from an airport, a restaurant, a car, an exhibition, and an AWGN. To get both clean and noisy signals, this research employed intermediate reference system (IRS) filters (*Loizou, 2009*). To achieve the appropriate SNR levels, the recovered noise segments were artificially introduced to the clean speech signal. The entire dataset was split into two sets: the training dataset and the testing dataset. 20% of the dataset was used for testing, while the remaining 80% was used to train the CNN algorithm (*Gholamy, Kreinovich & Kosheleva, 2018*).

- LibriSpeech

LibriSpeech noisy dataset consists of a single male voice with 6 different conditions under 3 different noises (*Panayotov et al., 2015*; *Park et al., 2020*). Three types of noises; Dog bark, Ambulance and fan noises, were examined at seven different SNRs: −10 dB, −5 dB, 0 dB, 5 dB, 10 dB, 15 dB, and 20 dB. 1 utterance from the LibriSpeech which is a clean speech signal was selected for training the CNN algorithm and 1 utterance with 18 different conditions was prepared with different noises at different levels of SNR. This entire dataset was used for both testing and training. 20% of the dataset was used to test the CNN algorithm and 80% of the dataset was used for training the CNN algorithm. We also considered this dataset to evaluate the performance accuracy of the proposed MCSE system in terms of spectrogram analysis and word recognition rate. This dataset is used to evaluate the performance of the proposed system.

The details of the AURORA and LibriSpeech datasets used in the experiments are presented in Table 3.

## The experimental setup

This research conducted two types of experiments namely the (1) benchmark MCSE based on beamforming, ANR and VAD algorithms (BAV-MCSE) and (2) the proposed MCSE based on deep learning algorithm with DWT pre-processing (DWT-CNN-MCSE).

## The benchmark MCSE system

In the benchmark experiment, the BAV-MCSE system is experimented with using Beamforming, ANR, and VAD algorithms (BAV). In this experiment, MEMS microphone array captures the noisy speech signals and sends the signals to fixed beamforming to separate the audio and noisy beams, the output of the beamforming goes to ANR to filter the noise based on reference noise, and finally, VAD separates the voiced speech signals and the result is evaluated with an ASR engine.

The experimental setup is as follows:

- Device configuration is based on the researchers' previous work *Cherukuru, Mumtaz Begum & Hema (2021)*

(a) Controller: stm32f103CBT6

(b) Main clock: 72MHz

(c) Memory: 128 KB ROM/20 KB RAM

**Table 3** Dataset used for experimenting multi-channel speech enhancement.

| Speech enhancement | Speech database | Training and test data | SNR/dB | Types of noises |
|---|---|---|---|---|
| The existing BAV-MCSE | AURORA | • 25 utterances of clean speech signals for training<br>• 42 sets of noise mixed signals used for testing | −10 dB, −5 dB, 0 dB, 5 dB, 10 dB, 15 dB | Airport, Babble, Car, Exhibition, restaurant and white Gaussian noise |
| The proposed DWT-CNN-MCSE | AURORA | • 80% of signals used for training to train the CNN model.<br>• 20% of signals used for testing | −10 dB, −5 dB, 0 dB, 5 dB, 10 dB, 15 dB | Airport, Babble, Car, Exhibition, restaurant and white Gaussian noise |
| | LibriSpeech | • 80% of signals used for training to train the CNN model.<br>• 20% of signals used for testing | −10 dB, −5 dB, 0 dB, 5 dB, 10 dB, 15 dB | Dog bark noise, Fan noise, Ambulance noise |

(d) External storage: Transcend 8 GB class 4 memory card.

(e) Transducers: capacitive electret microphones

(f) Servos: 9G servo

• Sampling setup

For sampling, we used pre-amplified two transducers output, which were passed to a single-stage bandpass filter (80 Hz–16 kHz), gain adjusted, level shifted to 1.75 V, and then fed to individual ADCs (analog to digital converter).

For this research, the ADCs were configured at 12-bit vertical resolution and 16,000 Samples per second (+/- 50). Data is saved in SD card using Conversion complete interrupt linked to the DMA channel that writes in SD card and Buffer variable defined in RAM, where both sampling times were synchronized. The amplifier used was LM358 general-purpose Opamp.

• Variability setup

Due to the increased ARM deflection of the Servos, the Timer 1 PWM channels were used to connect two 9G servos with 16-bit resolution. There is a 10 mm gap between each microphone.

• Setup of the noise and sample utterance system

Edifier 2.0, a channel speaker, served as the main noise maker. The noise samples were continuously looped and transferred from the BeagleBone Black Single Board to the amplifier, where the speech is altered. Only the left channel of the Logitech USB speakers were used to enter the samples into the Beagle Bone Black single-board computer.

• SNR setup

The required SNR ($-10$ dB, $-5$ dB, 0dB, 5 dB, 10 dB, 15 dB, and 20 dB) was derived by adjusting the noise sound amplifier gain and the sample utterance amplifier gain control.

## The proposed MCSE system

In the experiment, the proposed DWT-CNN-MCSE system is experimented using DWT as a pre-processing and CNN-based deep learning technique.

The proposed algorithm is implemented using MATLAB 2021a. This tool is widely adopted for various signal-processing tasks such as image processing, speech-processing, and ECG signals. DWT preprocessing algorithm is used to create the signal's detailed information by rescaling and iterating through a series of filters. Up-sampling and down-sampling (subsampling) processes determine the signal's resolution, whereas filtering operations determine its scale (resolution). In simple terms, DWT decomposes the signal into different frequency bands. It effectively denoises the speech signal and compresses the speech signal without any significant loss in speech quality. The output of the DWT goes to CNN in which data is processed through multiple layers such as convolution, pooling fully connected, max pooling, linear layer, BiLSTM, fully connected layer and softmax layer to learn its attributes and improve the recognition accuracy.

In this study, this research employed these tools for speech-processing tasks. The proposed algorithm uses the following toolboxes:

• Audio toolbox:

Audio Toolbox offers audio processing, speech analysis, and acoustic measurement tools. It provides algorithms to evaluate acoustic signal metrics, and to train machine learning and deep learning models. Researchers can import, categorise, and enhance audio data sets using Audio Toolbox.

• Data acquisition toolbox:

The Data Acquisition Toolbox$^{TM}$ includes programs and features for configuring data collection devices, reading data into MATLAB and Simulink, and publishing data to DAQ analogue and digital output channels.

• Digital signal processing toolbox

With the DSP System Toolbox, researchers can create and examine FIR, IIR, multi-rate, multistage, and adaptive filters.

- Wavelet toolbox

Wavelet Toolbox offers functions and applications to analyze and synthesize signals and images. Researchers can analyze signals and images at various resolutions using discrete wavelet analysis to find changepoints, discontinuities, and other events that are not immediately visible in raw data.

- Deep learning toolbox

With methodologies, pre-trained models, and applications, the deep Learning Toolbox was used for developing and integrating deep neural networks into applications.

## Evaluation methods

This research evaluates the performance of a multi-channel speech enhancement system in a noisy environment (stationary and non-stationary noise) using spectrogram analysis and WRR.

- Spectrogram analysis

The amplitude of speech signals is analyzed using spectrogram analysis (*Haykin et al., 1991*). MATLAB is used for time-domain spectrogram analysis for both stationary noises (white Gaussian noise) and non-stationary noises in the environment (Babble, Car, Exhibition, Airport, and Restaurant) from the AURORA database and ambulance noise from the LibriSpeech database. The spectrogram reflects the change in amplitude, frequency, wavelength and time at different levels of SNR's. In this experiment, we analyzed the spectrograms with signals amplitude in time domain.

- Word recognition rate (WRR)

The word recognition rate is used to assess the performance of multi-channel speech enhancement systems. WRR measures the performance accuracy of multi-channel speech enhancement system. The following formula is used to calculate WRR:

$$\text{Word Recognition Rate (WRR)} = 1 - \text{WER} \tag{1}$$

$$\text{Word Error Rate (WER)} = \frac{S + D + I}{N} \tag{2}$$

N is the total amount of words or letters in the sentence, S the number of times other words have been substituted for them, and D denotes the number of words that have been deleted. In a sentence, I represent the number of insertions.

## RESULTS

### Spectrogram analysis

Figure 4 shows the spectrograms of clean speech and Table IV shows and compares the spectrograms of a sample utterance under six different conditions from the AURORA database which include the spectrogram analysis: (a) Noisy speech at various SNR levels, and (b) enhanced speech applying benchmark BAV-MCSE (c) enhanced speech using

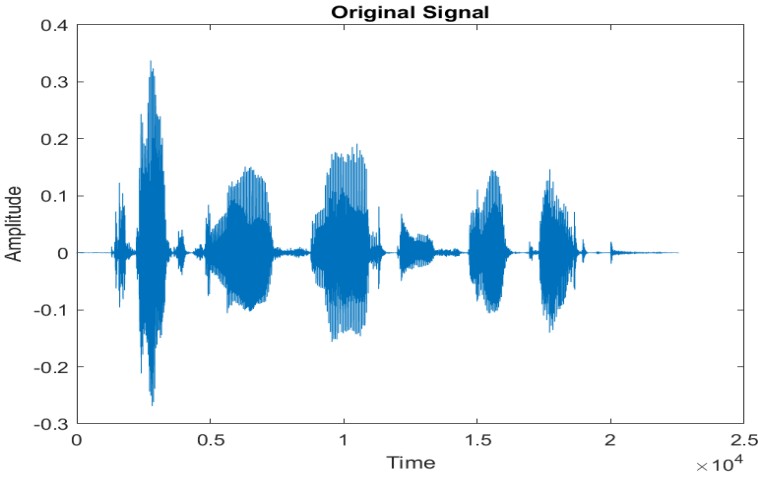

**Figure 4** **Clean speech signal.**

proposed DWT-CNN-MCSE for airport noise, babble noise, restaurant and white Gaussian noise.

The spectrogram of clean speech signal reflects the change in amplitude, wavelength and time and it represents high intelligibility and high word recognition rate. When this clean speech signals mixed with different types of noises such as airport, babble, car, exhibition, restaurant and ambulance noise, the changes in spectrograms of noisy speech signal in terms of amplitude and time domain differs compared to clean speech signal which is depicted in Table 4(a), (b), Table 5(a) and (b). After processing the noisy speech signals with the proposed method (DWT-CNN-MCSE), we observed the spectrograms of enhanced signals are close to the amplitude, wavelength and time of clean speech signals which are depicted in Tables 4(c) and 5(c).

The DWT-CNN-MCSE provides substantial noise suppression compared to unprocessed speech and the DWT-CNN algorithms eliminate almost all the noises in the entire frequency range.

This research has also experimented with the proposed DWT-CNN-MCSE using LibriSpeech. Table 5 shows the spectrograms of the original signal, noisy speech signal and enhanced speech signal under ambulance noise at −10 dB SNR level. As compared to ambulance original noisy speech signal, the proposed MCSE offers significant noise reduction, and the DWT-CNN algorithms nearly eliminate all noises over the whole frequency band.

## Word recognition rate (WRR)

Tables 6 and 7 present the results of the benchmark BAV-MCSE and proposed DWT-CNN-MCSE tested using the AURORA dataset at different levels of SNRs under stationary and non-stationary noisy environments. By comparing the performance of the developed noise reduction system in filtering various SNR of environmental noises, the following are

**Table 4** Spectrograms Analysis of Noisy Speech Signal and Enhanced Signal of AWGN, Airport, Babble, Car, Exhibition and Restaurant Noises.

| Noise at −5db SNR level | (a): Noisy Speech Signal | (b): Enhanced Signal/ Reconstructed Signal (BAV-MCSE) | (c): Enhanced Signal/ Reconstructed Signal (DWT-CNN-MCSE) |
|---|---|---|---|
| AWGN | | | |
| Airport | | | |
| Babble | | | |

**Table 4** (*continued*)

| Noise at −5db SNR level | (a): Noisy Speech Signal | (b): Enhanced Signal/ Reconstructed Signal (BAV-MCSE) | (c): Enhanced Signal/ Reconstructed Signal (DWT-CNN-MCSE) |
|---|---|---|---|
| Car | | | |

| Exhibition | | | |

| Restaurant | | | |

Cherukuru and Mustafa (2024), *PeerJ Comput. Sci.*, DOI 10.7717/peerj-cs.1901

**Table 5** **Spectrograms analysis of original, noisy speech signal and enhanced signal of ambulance noise.**

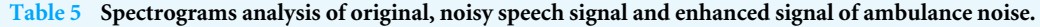

| Type of Noise at −5db | (a): Original Signal | (b): Noisy Speech Signal | (c): Enhanced speech signal (DWT-CNN-MCSE) |
| --- | --- | --- | --- |
| Ambulance Noise | | | |

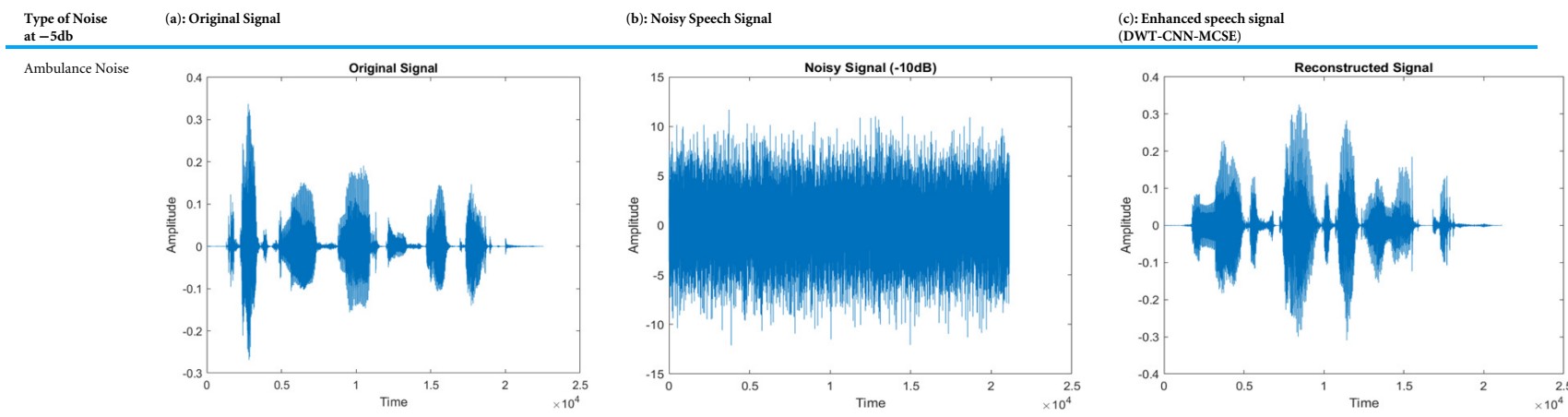

**Table 6  WRR Performance of benchmark MCSE and the proposed MCSE under non-stationary noises of the AURORA dataset.**

| Database | Noise | SNR/dB | BAV-MCSE WRR (%) | DWT-CNN-MCSE WRR (%) | DIFFERENCES WRR (%) |
|---|---|---|---|---|---|
| | | −10 | 5.82 | 70.55 | 64.73 |
| | | −5 | 12.32 | 72.51 | 60.19 |
| | Airport | 0 | 19.06 | 78.75 | 59.69 |
| | | 5 | 36.14 | 77.44 | 41.3 |
| | | 10 | 67.26 | 67.15 | −0.11 |
| | | 15 | 88.88 | 75.44 | −13.44 |
| | | −10 | 4.04 | 68.50 | 64.46 |
| | | −5 | 7.12 | 70.25 | 63.13 |
| | Babble | 0 | 17.56 | 70.32 | 52.76 |
| | | 5 | 35 | 66.44 | 31.44 |
| | | 10 | 74.18 | 73.79 | −0.39 |
| | | 15 | 90.64 | 61.49 | −29.15 |
| AURORA (Non-Stationary Noises) | | −10 | 7.26 | 72.50 | 65.24 |
| | | −5 | 13.26 | 74.60 | 61.34 |
| | Car | 0 | 16.55 | 80.49 | 63.94 |
| | | 5 | 35.16 | 77.44 | 42.28 |
| | | 10 | 67.2 | 81.49 | 14.29 |
| | | 15 | 92.02 | 78.8 | −13.22 |
| | | −10 | 6.54 | 65.15 | 58.61 |
| | | −5 | 11.54 | 68.25 | 56.71 |
| | Exhibition | 0 | 20.23 | 80.45 | 60.22 |
| | | 5 | 44.66 | 77.73 | 33.07 |
| | | 10 | 77.72 | 76.42 | −1.3 |
| | | 15 | 91.46 | 73.75 | −17.71 |
| | | −10 | 4.54 | 72.50 | 67.96 |
| | | −5 | 6.38 | 73.50 | 67.12 |
| | Restaurant | 0 | 12.12 | 80.39 | 68.27 |
| | | 5 | 38.24 | 77.75 | 39.51 |
| | | 10 | 55.56 | 76.35 | 20.79 |
| | | 15 | 75.2 | 74.48 | −0.72 |

obtained: a WRR of 70.55% at −10 dB SNR and 75.44% at 15 dB SNR, while 5.82% at −10 dB and 88.8% at 15 dB by the BAV-MCSE system.

For non-stationary noises, Fig. 5 demonstrates the variations in the WRR for both the BAV-MCSE and the proposed DWT-CNN-MCSE. In comparison to BAV-MCSE, the proposed MCSE is particularly good in recognizing speech in non-stationary noisy conditions. Finally, to determine if the results for BAV-MCSE and proposed DWT-CNN-MCSE differed significantly, we used Analysis of Variance (ANOVA), and the results are presented in Fig. 6. From Fig. 6, the result shows that the proposed algorithms' scores are significantly different from the existing algorithm (BAV-MCSE) under non-stationary

**Table 7 WRR performance of the benchmark MCSE and the proposed MCSE under stationary noises of the AURORA dataset.**

| Database | | SNR/dB | BAV-MCSE WRR (%) | DWT-CNN-MCSE WRR (%) | Difference WRR (%) |
|---|---|---|---|---|---|
| AURORA | Stationary Noise: Additive white Gaussian noise (AWGN) | −10 | 42.6 | 74.48 | 31.88 |
| | | −5 | 58.3 | 72.51 | 14.21 |
| | | 0 | 63.4 | 73.55 | 10.15 |
| | | 5 | 68.5 | 79.78 | 11.28 |
| | | 10 | 72.6 | 75.79 | 3.19 |
| | | 15 | 74.08 | 74.38 | 0.3 |

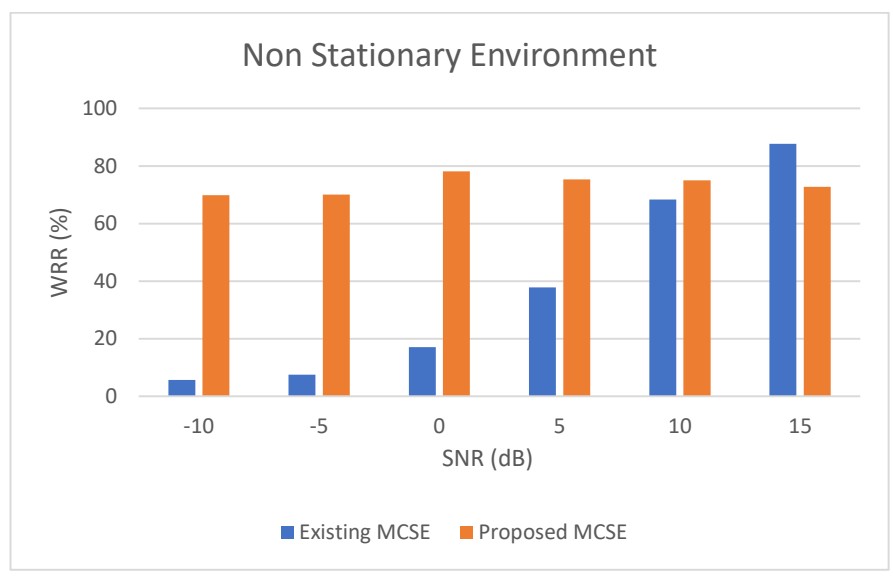

**Figure 5 WRR of existing BAV-MCSE *Vs* proposed DWT-CNN-MCSE under non- stationary environment.**

environment. This further reveal that the proposed algorithm (DWT-CNN-MCSE) has statistically different result at *p*-value less than 0.05.

For stationary noises, Fig. 7 demonstrates the changes in the WRR for both the BAV-MCSE and the proposed DWT-CNN-MCSE. In comparison to the existing BAV-MCSE, the proposed DWT-CNN-MCSE is good in recognizing speech in stationary noisy conditions. Finally, to determine if the results for BAV-MCSE and proposed DWT-CNN-MCSE were significantly different, we used Analysis of Variance (ANOVA), and the results are depicted in Fig. 8. From Fig. 8, the result shows that the proposed DWT-CNN-MCSE scores are significantly different from the existing BAV-MCSE under stationary environment at *p* value less than 0.05.

Table 8 presents the results of the proposed DWT-CNN-MCSE tested using the LibriSpeech dataset at different levels of SNRs for the ambulance, dog bark and fan

ANOVA

| Source of Variation | SS | Df | MS | F | P-value | F crit |
|---|---|---|---|---|---|---|
| Between Groups | 3925.964225 | 1 | 3925.964225 | 6.707204873 | 0.026957951 | 4.964602744 |
| Within Groups | 5853.353669 | 10 | 585.3353669 | | | |
| Total | 9779.317895 | 11 | | | | |

**Figure 6**  **Results of ANOVA.**

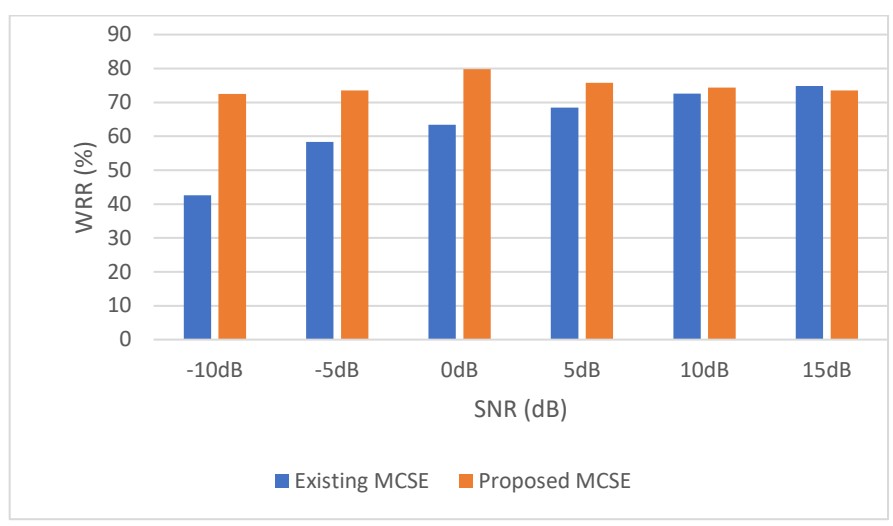

**Figure 7**  **WRR of the existing BAV-MCSE *vs* the proposed DWT-CNN-MCSE systems under stationary environment.**

ANOVA

| Source of Variation | SS | Df | MS | F | P-value | F crit |
|---|---|---|---|---|---|---|
| Between Groups | 400.092 | 1 | 400.092 | 5.453223 | 0.041678 | 4.964603 |
| Within Groups | 733.6799 | 10 | 73.36799 | | | |
| Total | 1133.772 | 11 | | | | |

**Figure 8**  **Results of ANOVA.**

**Table 8  WRR performance of proposed WSE under non-stationary noises.**

| Database | Noise | SNR/DB | WRR |
|---|---|---|---|
| | | −10 dB | 65.55 |
| | | −5 dB | 63.25 |
| | Dog bark Noise | 0 dB | 74.38 |
| | | 5 dB | 70.68 |
| | | 10 dB | 67.45 |
| | | 15 dB | 66.58 |
| | | −10 dB | 65.55 |
| | | −5 dB | 63.25 |
| LibriSpeech | Ambulance Noise | 0 dB | 72.88 |
| | | 5 dB | 70.65 |
| | | 10 dB | 66.45 |
| | | 15 dB | 65.48 |
| | | −10 dB | 62.20 |
| | | −5 dB | 61.25 |
| | Fan Noise | 0 dB | 73.89 |
| | | 5 dB | 69.88 |
| | | 10 dB | 67.86 |
| | | 15 dB | 66.75 |

noises. We noticed that among all the three noises, the highest WRR of 74.38% is obtained at 0dB under dog bark and lowest WRR of 61.25% is obtained at −5 dB under fan noise.

## DISCUSSION

In this study, we carried out spectrogram analysis and WRR on MCSE systems. The spectrograms of noisy speech signals and enhanced speech signals of different noisy speech signals at −5 dB SNR from the AURORA dataset are presented for both the benchmark and proposed MCSE system. As compared to the benchmark system, the proposed system's enhanced spectrograms have clear signals and are closer to the clean speech signals under stationary and non-stationary environmental noise. Similarly, for the LibriSpeech dataset, spectrograms of noisy speech signals and enhanced speech signals were analyzed and the proposed MCSE showed a clear signal and closer to the clean signal under ambulance noises. We noticed the spectrograms of the enhanced speech signals by DWT-CNN-MCSE offer significant noise reduction when compared to raw noisy speech, and the DWT-CNN algorithms filtered maximum noise throughout the whole signal spectrum.

The spectrograms of the enhanced speech obtained with all processing methods are depicted in Tables 4 and 5. The spectrograms of BAV-MCSE have lost some important speech contents such as some of the speeches are missing, hence provided less speech recognition rate as compared to DWT-CNN-MCSE which is evident in Tables 4(c) and 5(c). If we note the spectrogram of DWT-CNN-MCSE, we obtained a close replica of the clean speech spectrogram and important speech contents are effectively preserved. Also, low noise is observed in the spectrogram of DWT-CNN-MCSE output speech. The

time-domain waveforms of the enhanced speech utterances obtained with all the processing methods are depicted in Tables 4 and 5. The waveforms of BAV-MCSE have some noise, hence provided less recognition rate as compared to DWT-CNN-MCSE which is evident in Tables 4(b), (c) and 5(b), (c). Low noise is observed in the waveform of DWT-CNN-MCSE output speech.

The result of word recognition shows that the benchmark MCSE could not function adequately in low SNR settings. However, MCSE performed better in a noisy stationary environment than non-stationary environment. It was also discovered that the MCSE algorithms perform well in both stationary and nonstationary noisy environments at high SNR. The linear relationship between SNR and WRR shows that MCSE successfully filters noise at higher SNR and not at lower SNR, as the strength of the noise is too low for MCSE to filter it out as beamforming, ANR and VAD algorithms are more sensitive at low SNR conditions.

We also analyzed the proposed system in terms of WRR at different levels of SNR under stationary and non-stationary noises from the AURORA and LibriSpeech databases. The result of the WRR shows MCSE could function adequately in low and high SNR conditions under stationary and non-stationary noisy environments. However, there is a need to make the DWT-CNN-MCSE robust noisy free system.

The proposed framework in MCSE, which included a pre-processing algorithm based on DWT and a deep learning algorithm based on CNN, outperformed the benchmark algorithms in detection in noisy environments, especially at low SNR conditions in terms of word recognition rate accuracy. We noticed that WRR of the benchmark MCSE provided good results only at 15dB SNR compared to the proposed MCSE. It shows that the proposed MCSE is sensitive at 15dB SNR level under non-stationary environments. We also noticed that the proposed MCSE has a WRR that is twice as high as the benchmark MCSE at −10 dB, −5 dB, 0 dB, and 5 dB SNR levels under both stationary and non-stationary environments.

## CONCLUSION

The multi-channel speech activity-related devices are commonly used in various real-time applications, and the communication or speech quality performance of these devices is degraded by various types of noises. Thus, improving the quality of speech signals is important for these multi-channel devices. To deal with various environmental noises, we propose an MCSE using deep learning and preprocessing algorithms and examine the performance of the proposed MCSE system in filtering the environmental noises at low to high SNR. A new architecture is presented, which considers wavelet transform, deep learning (CNN), and BiLSTM model to learn the data pattern and trained to obtain the filtered signals. The proposed system shows considerable performance when compared to related studies. By comparing the performance of the proposed system in handling various SNRs of environmental noises, it achieved a WRR of 70.55% at −10 dB SNR and 75.44% at 15 dB SNR, as compared to the existing MCSE system at 5.82% at −10 dB and 88.8% at 15 dB. It can be inferred from the comparison that the proposed system has outperformed

the benchmark MCSE system. From the ANOVA analysis, the result indicated that the proposed MCSE's scores are significantly different from the existing MCSE system. The word recognition accuracy is achieved at an acceptable rate at low SNR.

## FUTURE DIRECTIONS

Our current work focuses on deep learning algorithms, where the proposed system outperforms the benchmark system. However, one of the limitations of deep learning-based algorithms is the high computational costs. As such, implementing deep learning-based approaches in portable communication devices may be difficult due to the low computing power of these devices. In this regard, one possible way is to combine the existing filters and deep learning approaches to enhance the speech quality and intelligibility of the output. We plan to conduct more investigations into wider types of noises and more effective speech enhancement algorithms to improve the performance of the multi-channel speech enhancement system.

### Funding

This research was financially supported by Ministry of Higher Education under the Fundamental Research Grant Scheme (FRGS/1/2023/ICT09/UM/02/1 and FRGS/1/2020/ICT09/UM/02/1). The funders had no role in study design, data collection and analysis, decision to publish, or preparation of the manuscript.

### Grant Disclosures

The following grant information was disclosed by the authors:
Ministry of Higher Education under the Fundamental Research Grant Scheme: FRGS/1/2023/ICT09/UM/02/1, FRGS/1/2020/ICT09/UM/02/1.

### Competing Interests

The authors declare there are no competing interests.

### Author Contributions

- Pavani Cherukuru conceived and designed the experiments, performed the experiments, analyzed the data, performed the computation work, prepared figures and/or tables, authored or reviewed drafts of the article, and approved the final draft.
- Mumtaz Begum Mustafa conceived and designed the experiments, performed the experiments, analyzed the data, performed the computation work, prepared figures and/or tables, authored or reviewed drafts of the article, and approved the final draft.

### Data Availability

  The code used in this study are available in the Supplemental Files.
  The AURORA database is available at NOIZEUS: A noisy speech corpus for evaluation of speech enhancement algorithms:

https://ecs.utdallas.edu/loizou/speech/noizeus/.

## Supplemental Information

Supplemental information for this article can be found online at http://dx.doi.org/10.7717/peerj-cs.1901#supplemental-information.

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
