# Peer review of "CNN-based noise reduction for multi-channel speech enhancement system with discrete wavelet transform (DWT) preprocessing"

_PeerJ Computer Science, doi:10.7717/peerj-cs.1901_

## Round 0.1 · original submission · Major Revisions

I urge you to please carry out the revisions as recommended by the reviewers & resubmit the manuscript. Reviewer 1 has submitted a separate document with his/her comments and suggestions. In my opinion, these will result in a significantly enhanced manuscript in terms of narration, presentation, and overall contributions.

**Language Note:** The review process has identified that the English language must be improved. PeerJ can provide language editing services - please contact us at copyediting@peerj.com for pricing (be sure to provide your manuscript number and title). Alternatively, you should make your own arrangements to improve the language quality and provide details in your response letter. – PeerJ Staff

Reviewer 1 ·

Basic reporting

The paper is well written and structured but needs changes, please see comments file.

Experimental design

Please see the comments. This section needs changes I have mentioned in the comments file.

Validity of the findings

Please see the comments file.

Additional comments

My recommendations to accept the paper after addressing the above comments.

Annotated reviews are not available for download in order to protect the identity of reviewers who chose to remain anonymous.

·

Basic reporting

no comment

Experimental design

The Following points should be included by author for the revision.
• While the study mentions that the existing MCSE system was tested only for White Gaussian noise, the types of non-stationary environmental noises tested in this study aren't specified in detail.
• There's a lack of deeper explanation on the technical workings of the DWT-CNN-MCSE system, such as the CNN architecture and specifics of the DWT preprocessing, which could benefit readers looking for technical insights.
• The research doesn't seem to factor in real-world challenges like microphone placement, distance from the sound source, or overlapping speech.
• Environmental noises pose significant challenges to speech enhancement systems, and their variability necessitates systems that can adapt to both stationary and non-stationary noises.
• The DWT-CNN-MCSE system's use of wavelet transforms and neural networks presents a promising approach to tackle the challenges of low SNRs, showing marked improvement over the BAV-MCSE approach.
• The varied performance across different SNR levels underscores the importance of designing speech enhancement systems that can dynamically adapt to different noise levels.

Validity of the findings

no comment

Additional comments

Suggestions:

• A deeper dive into the DWT and CNN components of the proposed system would provide readers with a better understanding of its mechanics and potential advantages.
• Simulating real-world conditions, like fluctuating distances between speakers and microphones, or handling overlapping speech, can provide insights into the system's practical applicability.
• Exploring the performance of the proposed system beyond the -10dB to 20dB range could demonstrate its resilience in more extreme noisy situations.

---

## Round 0.2 · Minor Revisions

Please address the remaining comments from Reviewer 1, as per their original review.

Reviewer 1 ·

Basic reporting

The revised version has suffecient basic reporting now. No further comments on this section.

Experimental design

Experimental design has been improved, suggested experiments are added or further improved.

Validity of the findings

The authors have improved the paper after performing further research.

Additional comments

Although all comments are addressed,
Comment 2 seems to be pending.
Comment 8-9 needs some more explanation.

·

Basic reporting

no comment

Experimental design

no comment

Validity of the findings

no comment

Additional comments

As all my comments were more or less properly addressed by the authors in the revised version of the paper, I am pleased to accept the paper for a publication.

---

## Round 0.3 · accepted · Accept

Thank you for addressing the final round of minor revisions. In light of the reviewer's positive reassessment and my own check of the new revision, I'm happy to recommend the paper for acceptance.

Reviewer 1 ·

Basic reporting

This is a revised version of the initial submission. As a reviewer of this manuscript, I have raised various questions in the first round that are properly addressed by the authors. I appreciate the efforts of the authors. I have no further comments on this submission. My recommendation is to accept the paper in its current version. Thank you

Experimental design

Yes

Validity of the findings

Yes

Additional comments

NA